The effects of naproxen sodium on the growth, reproduction, survival, and feeding of a freshwater pond snail

Becker Rachel rachelbecker1109@gmail.com
Hamman Elizabeth
Department of Biology, St. Mary’s College of Maryland , St. Mary’s City , MD , United States of America
Oehlmann Jörg
Electronic publication date: 2025 Oct 14
Publication date: 2025
Volume: 13
Electronic Location ID: e20163
Received 2024 Dec 4; Accepted 2025 Sep 10
Copyright: ©2025 Becker and Hamman
Copyright year: 2025
Copyright holder: Becker and Hamman
License: This is an open access article distributed under the terms of the Creative Commons Attribution License, which permits unrestricted use, distribution, reproduction and adaptation in any medium and for any purpose provided that it is properly attributed. For attribution, the original author(s), title, publication source (PeerJ) and either DOI or URL of the article must be cited.
License URL: https://creativecommons.org/licenses/by/4.0/

Keywords: Pharmaceutical pollution, NSAIDs, Physa acuta, Naproxen sodium, Growth, Survival, Reproduction

Funding: St. Mary’s College of Maryland St. Mary’s College of Maryland funded this study through St. Mary’s Project funds awarded to Rachel Becker. The funders had no role in study design, data collection and analysis, decision to publish, or preparation of the manuscript.

==============================
Over-the-counter drugs are emerging contaminants in the environment. These contaminants often affect aquatic communities, including freshwater invertebrates. Nonsteroidal anti-inflammatory drugs (NSAIDs) are commonly found in aquatic ecosystems worldwide and have documented negative effects on aquatic organisms. Naproxen is often suggested as a more environmentally friendly alternative due to less bioconcentration in fish. However, studies have yet to test the effects of naproxen on freshwater molluscs. This study exposed pond snails, Physa spp., to three nominal levels of naproxen sodium, 100 µg/L, 1,000 µg/L and 10,000 µg/L, along with a control, over a month-long experiment. We measured snail growth, survival, reproduction, and feeding. Naproxen reduced snail survival across all levels of naproxen exposure and the rate that snails fed at the highest level of naproxen. However, we did not detect an effect of naproxen on snail growth or reproduction rates, even at pollution levels well above those observed in natural systems. Naproxen reduced the grazing of Physa spp. only at our highest concentration, but reduced survival by at least 50% across all concentrations.

Introduction

Pharmaceutical pollution is widespread, with more than 600 pharmaceuticals documented in natural environments across all seven continents (aus der Beek et al., 2016). Aquatic environments are particularly susceptible to pharmaceutical pollution (Cizmas et al., 2015; Ebele, Abou-Elwafa Abdallah & Harrad, 2017; Adeleye et al., 2022), as pharmaceuticals are excreted post-consumption into wastewater. Insufficient wastewater treatment methods cause chemicals to enter the natural environment (Deblonde, Cossu-Leguille & Hartemann, 2011). As a result, freshwater organisms are exposed to a large variety of pharmaceuticals at biologically relevant concentrations, the most common of which are antibiotics, antidepressants, and nonsteroidal anti-inflammatory drugs (NSAIDs) (Świacka et al., 2022).

NSAIDs include painkillers and anti-inflammatories such as ibuprofen, aspirin, diclofenac, and naproxen. These drugs, including naproxen, are commonly found at concentrations of over 100 ng/L in surface waters across the globe (Ebele, Abou-Elwafa Abdallah & Harrad, 2017; Wojcieszyńska & Guzik, 2020), and their pollution negatively affects occupants of aquatic habitats by altering the reproductive capacity, growth rates, and survival of multiple organisms (Świacka et al., 2022). For example, diclofenac reduces egg production in crustaceans, including Daphnia magna, Thamnocephalus platyurus, Ceriodaphnia dubia, and Hyalella azteca (Du et al., 2016; Parolini, 2020). It also increases mortality in juvenile brown trout (Salmo trutta) (Schwarz et al., 2017). One of the most commonly used NSAIDs, ibuprofen, has a teratogenic effect on carp (Cyprinus carpio) that results in delayed hatching and yolk deformation in embryonic development (Gutiérrez-Noya et al., 2020). Because changes in demographic processes, such as birth rates or mortality, can have large effects on population size, understanding the effects of NSAIDs on these rates is critical to predicting how increased pollution will affect ecological populations and communities (Brown, Bernot & Bernot, 2012).

One less commonly studied, yet commonly used NSAID is naproxen (Moreno Ríos et al., 2022). Naproxen reduces the survival and growth of the crustacean Hyalella azteca (Lucero et al., 2015), the medaka fish (Oryzias latipes) (Kwak et al., 2018), and the rotifer Brachionus calyciflorus (Isidori et al., 2005; Kwak et al., 2018; Parolini, 2020). It also negatively affects multiple organs, including the thyroid, liver, and heart in zebrafish (Danio rerio) (Li et al., 2016; Xu et al., 2019), and the kidney, liver, and heart of the three-spined stickleback (Gasterosteus aculeatus) (Näslund et al., 2020). Additionally, the effects of naproxen are not limited to the chemical itself. When toad tadpoles (Anaxyrus terrestris) were exposed to the two phototransformation products of naproxen, they showed greater negative effects than when exposed to naproxen itself (Cory et al., 2019).

Despite these harmful effects, some authors posit that naproxen might be a preferable substitute for other NSAIDs, such as diclofenac, due to lower rates of bioconcentration (Näslund et al., 2020; Parolini, 2020). Therefore, it is particularly troubling that we lack information on the effects of naproxen on the demographic rates (e.g., reproduction and survival rates) of key freshwater organisms, such as molluscs.

Molluscs, particularly snails, have been found to be good bioindicators (e.g., for heavy metal pollution, Mahmoud & Abu Taleb, 2013) and are often used to test the effects of contaminants. They are also a good indicator of how unsafe a certain contaminant is for the overall ecosystem (Barky et al., 2012; Banaee & Taheri, 2019). In fact, a 2022 review of pharmaceuticals in aquatic organisms found molluscs second to fish as the focal organism in studies of pharmaceutical exposure (Świacka et al., 2022). Freshwater snails are susceptible to NSAID pollution. Ibuprofen lowers the hatching rates of Planorbis carinatus (Pounds et al., 2008), and exposure to diclofenac reduces Lymnaea stagnalis’s growth and feeding rates (Bouly et al., 2022a), along with causing weakened immune responses (Boisseaux et al., 2017). In some cases, the magnitude of the effect depends on the generations that are exposed (Bouly et al., 2022b).

While some NSAIDs, such as ibuprofen and diclofenac, are somewhat well-studied in molluscs, others, such as naproxen, are much less so, yet naproxen has been found in their tissue (Wolecki et al., 2019), and they might be susceptible to the harmful effects as seen with other organisms. To our knowledge, the impact of naproxen on the growth, reproduction, survival, and feeding of freshwater snails has not been previously studied.

Physid snails are commonly found in freshwater habitats across their native range of North America (Lydeard, Campbell & Golz, 2016) as well as invasive ranges, including Europe (Van Leeuwen et al., 2013), South Africa (De Kock & Wolmarans, 2007), and Chile (Collado et al., 2020). This snail is susceptible to the effects of pollution from pharmaceutical and personal care products. For example, the snail has reduced growth and locomotion in response to triclosan, an antimicrobial agent (Brown, Bernot & Bernot, 2012), decreased movement in response to acetaminophen (Elias et al., 2025a), and increased mortality in response to personal care product pollution (Sobrino-Figueroa, 2015). The snail has been noted as an indicator species for heavy metal pollution (Mahmoud & Abu Taleb, 2013), making it an excellent model for freshwater toxicological research, yet understudied with regard to NSAIDs.

Given the need for more information on naproxen on freshwater invertebrates, and of NSAIDs on physid snails, we tested the effects of variable concentrations of naproxen on the demographic and ecological rates of the freshwater pond snail, Physa spp. We predicted that higher concentrations of naproxen would reduce growth rates and feeding frequency, as well as increase mortality and reduce egg sac production.

Materials and Methods

Animal housing and care

We purchased pond snails (Physa spp.) from Carolina Biological Supply Co. and held them in a temperature-controlled room at 23 °C with LED lights on a 12-hour cycle. Snails were fed Aquatic Arts Algae Wafers every 3 days (per food manufacturer directions) and their water was changed biweekly from premade solutions. Initially, snails were held in a tank of dechlorinated tap water, and then transitioned to Holtfreter’s solution over the first week. This solution was made of distilled water, sodium chloride (3.50 g/L), calcium chloride dihydrate (0.133 g/L), magnesium sulfate heptahydrate (0.205 g/L), all procured from Fisher Scientific, and sodium bicarbonate (Arm and Hammer, 0.200 g/L). We also supplemented with calcium carbonate (98%, Fisher Scientific, Hampton, NH, USA). We then allowed the snails to further acclimate in the tank for another week prior to the start of the experiment, where they were placed in individual glass mason jars filled to 0.3L with their experimental treatment solution, and the experiment began upon exposure.

Experimental methods

To test the effect of naproxen on the snail demographic rates (reproduction, growth, and survival), we conducted a month-long lab experiment where snails were housed in individual containers (one snail per jar) with four nominal levels of naproxen sodium (Fisher Scientific, Hampton, NH, USA) exposure in Holtfreter’s solution: 0 µg/L, 100 µg/L, 1,000 µg/L, and 10,000 µg/L, with 10 replicates for each treatment (n = 10). Naproxen is often sold in the form of naproxen sodium, as the salt version is easier to absorb (Brutzkus, Shahrokhi & Varacallo, 2025), and is often used to test the biological effects of naproxen in non-target organisms (e.g., Li et al., 2016; Kwak et al., 2018; Cory et al., 2019). Although higher than currently observed concentrations (107 ng/L, Ebele, Abou-Elwafa Abdallah & Harrad, 2017), our experimental levels (100–10,000 µg) reflect the range from prior studies of naproxen on fish (Ji et al., 2013) to levels in line with those found in human urine (Aresta, Palmisano & Zambonin, 2005). We were unable to verify the nominal concentrations, although previous studies note that naproxen is fairly stable in water over short time periods (Ji et al., 2013; Xu et al., 2019), with other studies finding a half-life of over one year (Toński et al., 2019).

Each level of naproxen had 10 replicates for each treatment (n = 10) for a total of 40 snails. Snails were haphazardly assigned to a jar, and each jar was randomly assigned to a treatment. The jars were randomly arranged in space in the temperature control room

We measured growth, reproduction, and survival weekly and feeding every 3 days. To measure growth, we measured the length of the snail (apex to base) with calipers and calculated a weekly growth rate (change in length). Physa spp. are hermaphrodites and will self-fertilize, so we measured reproduction by recording the number of egg sacs produced each week. Finally, we noted whether each snail was living or dead.

To measure the effect of naproxen on snail feeding, we placed 2 g of an algae aquarium food wafer (Aquatic Arts, Indianapolis, IN, USA) in each jar for four hours. At the end of the feeding period, we noted the presence or absence of a grazing scar.

Statistical analyses

We used a series of analyses to test the effects of naproxen concentration on snail demographics and feeding. We used generalized linear mixed effects models implemented in “glmmTMB” (Brooks et al., 2017) for snail growth, reproduction, and feeding. Only living snails were included in the analysis of growth, reproduction, and feeding. Because we were unable to verify the nominal concentration, we treated this value as a categorical variable, rather than a continuous concentration. For all models, week (continuous) and treatment (factor) were fixed effects along with their interaction and the individual snail was a random effect. We evaluated all models using the “DHARMa” package (Hartig, 2022), which included evaluating overdispersion. The model for snail growth assumed a Gaussian distribution and the validation process indicated a quadratic term on time was necessary for an adequate fit. Snail reproduction (number of egg sacs) assumed a Poisson distribution due to the count data, and feeding a binomial distribution (food consumed or not).

To analyze the survival data, we used a Cox proportional hazards model implemented in the “survival” package (Therneau, 2024), with post hoc analysis using the “survminer” package for pairwise comparisons (Kassambara, Kosinski & Biecek, 2016). All analyses were performed in R version 4.4.1 (R Core Team, 2024).

Results

Throughout the experiment, snail length increased at a rate of approximately 0.55 mm/week, although the growth rate declined throughout the experiment (Fig. 1A, χ2(1) = 74.2, p < 0.001). Most growth occurred during the first week (an average of 1.41 mm across all snails), with only an average of 0.25 mm/week occurring during weeks two through four. The growth rate was not significantly affected by the level of naproxen exposure (χ2(3) = 2.8, p = 0.421), nor was there a detectable effect on the change in growth rate (interaction between week and treatment) throughout the experiment (χ2(3) = 0.768, p = 0.857)

Figure 1 Effect of naproxen sodium on Physa demographic rates.

(A) Weekly snail growth rate (change in length from apex to tip) through time, (B) the number of egg sacs produced by individual snails, and (C) survival rates. In (A) and (B), the lines indicate the median, the boxes indicate the first and third quartiles, and the whiskers 1.5 times the interquartile range. The error bars in (C) indicate +/- SE. Neither growth nor reproduction were significantly affected by exposure to naproxen, but survival rates were decreased at all three exposure concentrations relative to the control.

Snail egg production decreased throughout the experiment, but this change was not significant (Fig. 1B, χ2(1) = 3.67, p = 0.553). At the end of the first week, snails produced an average of 0.88 egg sacs across all treatments. However, at the conclusion of the experiment, snails only produced an average of 0.4 egg sacs. Similarly to growth rate, there was no detectable effect of naproxen by treatment (χ2(3) = 5.03, p = 0.169), nor the change in egg production (interaction between week and treatment) throughout the experiment (χ2(3) = 0.754, p = 0.860)

All snails survived the first week of the experiment, and all control snails survived the first two weeks (Fig. 1C). At the conclusion of the experiment, 15 out of 40 snails remained (7, 3, 3, and 2 for 0 µg/L, 100 µg/L, 1,000 µg/L, and 10,000 µg/L of nominal naproxen exposure, respectively). The Cox model indicated a significant effect of treatment on snail survival (χ2(3) = 10.31, p = 0.016), and pairwise comparisons showed a harmful effect of all levels of naproxen on the survival of Physa spp. compared to the control (p = 0.051, p = 0.051, and p = 0.016 for 100 µg/L, 1,000 µg/L, and 10,000 µg/L of naproxen, respectively), but no differences between 100 µg/L and 1000 µg/L (p = 0.896), 100 µg/L and 10,000 µg/L (p = 0.396), or 1,000 µg/L and 10,000 µg/L (p = 0.370).

Approximately 71% of snails ate each week across all treatments, with a slight decrease throughout the experiment (Fig. 2, χ2(1) = 7.33, p = 0.007). The level of naproxen exposure did affect snail feeding (χ2(3)=14.3, p = 0.002). Based on posthoc pairwise comparisons, snails exposed to the highest level of naproxen (10,000 µg/L) fed 41% less than snails in the other three treatments (p = 0.0087, p = 0.0387, and p = 0.0095 compared to the 0 µg/L, 100 µg/L, and 1,000 µg/L, respectively), while there was no significant difference between the control and 100 µg/L (p = 0.956), the control and 1,000 µg/L (p = 0.999), nor between 100 µg/L and 1,000 µg/L (p = 0.939).

Figure 2 Effect of naproxen sodium on Physa feeding rates.

Percentage of snails that ate food by treatment over time +/- SE. Snails exposed to the highest level of naproxen showed reduced feeding compared to all other treatments.

Discussion

Naproxen, an over-the-counter NSAID commonly used as a painkiller, is a common aquatic pollutant (Ebele, Abou-Elwafa Abdallah & Harrad, 2017). However, it might be preferable to other, more harmful NSAIDs, such as diclofenac, which has wide-reaching ecological and physiological effects (Näslund et al., 2020). In our experiment, we did not detect an effect of naproxen on growth (Fig. 1A) or reproduction (Fig. 1B), but naproxen did reduce survival (Fig. 1C). Additionally, snails exposed to our highest concentration (10,000 µg/L) of naproxen were less likely to feed than snails with lower levels of exposure (Fig. 2).

We found that at all tested levels of naproxen, the survival of Physa was reduced. By week 3, all exposure treatments were below a 50% survival rate, indicating that naproxen pollution could have large consequences for snail population sizes. Physid snails affect other members of the freshwater community through trophic interactions as prey for flatworms such as Dugesia lugubris (Tripet & Perrin, 1994), crayfish such as Procambarus simulans (Alexander & Covich, 1991), and fish such as Lepomis macrochirus. While much of this research has focused on the effects predators have on the snails (e.g., Goeppner et al., 2020), removing a key prey species has the potential to shift dynamics within the community.

Reductions in grazing due to naproxen pollution, either because of lower feeding or survival rates, could have cascading effects on the floral community and nutrient cycling (Bernot, 2013). Freshwater snails have large effects on their community through their grazing. Snail grazing reduces the amount and diversity of periphyton communities (Lowe & Hunter, 1988; Rezanka & Hershey, 2003) and the growth of macrophytes (Yang et al., 2020a). Given that predator exclusions can alter the periphyton community due to intense snail grazing (Martin et al., 1992), it is likely that reducing snail grazing due to pollutant presence would also affect community composition.

Many additional factors can affect the grazing behaviors of freshwater snails, such as parasite presence (Bernot & Lamberti, 2008), sediment loading (Kent & Stelzer, 2008), predator identity (Bernot & Turner, 2001), and water flow (Schössow, Arndt & Becker, 2016). In natural systems, snails will be subject to these factors in combination with pharmaceutical pollution, which could yield unpredictable results when pharmaceutical pollution can alter species interactions such as predation (Bose et al., 2022). Therefore, future work should include not only naproxen, but also naproxen in combinations with other abiotic and biotic stressors of Physa spp.

Additionally, while we were unable to detect an effect of naproxen on reproduction, we only counted egg masses. It is possible that instead of limiting the number of egg sacs produced, naproxen limits the hatching rate, the number of eggs in each sac, and/or affects the egg size. Reductions in hatching rates of Physa have been observed with other pollutants such as microplastics (Michler-Kozma, Neu & Gabel, 2022; Merbt et al., 2024). If this is also the case for naproxen and Physa, there could be additional negative ramifications on snail population dynamics, especially given the harmful effects of naproxen on survival.

One effect that was consistent throughout multiple response variables was the deterioration in snail health as the experiment progressed. Even with consistent feeding rates (Fig. 2), water changes, and calcium carbonate supplementation, snail demographic rates decreased throughout the experiment (Fig. 1), suggesting less than favorable conditions in their artificial habitats. Our experiment lasted longer than many studies on contaminants and Physa survival (e.g., 96 h (Gao et al., 2017)), and it is possible these small containers are not ideal for longer-term experiments. Given this result, it is possible the effects of naproxen were underestimated in this study due to this additional variability or overestimated if the effects of naproxen were exacerbated by laboratory conditions.

Pharmaceutical pollution can have large consequences on aquatic ecosystems (e.g., affect the recovery of a community after a disturbance, (Michelangeli et al., 2024), as well as neighboring terrestrial habitats (Previšić et al., 2021). The degree to which those effects propagate throughout the ecosystem can depend on the effects on freshwater molluscs. Pharmaceuticals can be passed up the food chain through freshwater snails (Yang et al., 2020b), similarly to how they transfer other pollutants, such as microplastics (Holzer et al., 2022), although the degree of transfer depends on the pharmaceutical. Previous studies have proposed naproxen as a preferred NSAID because it bioaccumulates less than alternatives such as diclofenac (Näslund et al., 2020)

Previous studies found harmful effects of NSAIDs such as ibuprofen (Pounds et al., 2008; Elías et al., 2025b), diclofenac (Bouly et al., 2022a) on mollusc growth. While we failed to detect a significant effect, our results (Fig. 1A) also trended in this direction. Some studies have noted complexities in the response of growth to NSAID exposure (Bouly et al., 2022b; Elías et al., 2025b), indicating the need to explore exposure over the entire development period, multiple generations, and in the presence of other contaminants. Variation also exists in the literature regarding the response of mollusc reproduction to NSAID exposure. While ibuprofen actually increased snail reproduction (Elías et al., 2025b), diclofenac, like our findings with naproxen (Fig. 1B), had no effect (Bouly et al., 2022a).

We found a harmful effect of naproxen on survival (Fig. 1C), which aligns with the effects of ibuprofen (Pounds et al., 2008). However, not all studies found a harmful effect (e.g., Bouly et al., 2022a). Given these disparate findings, it is likely that not only context and concentration, but the specific species play an important role in the response of freshwater molluscs to NSAID pollution. Future work is needed to compare the response of snails and other freshwater organisms to other NSAIDs to better understand the effects of NSAID pollution on freshwater communities.

Conclusions

Our study tested how exposure to high levels of naproxen, an NSAID and emerging freshwater contaminant, affected the growth, reproduction, survival, and feeding of Physa spp. snails. Naproxen did not affect growth or reproduction, but did reduce survival at all levels of exposure and feeding at the highest level. Because freshwater snails are important to freshwater food webs, future studies should extend the range of naproxen exposure and explore the effects of other NSAIDs to better understand the effects of these pollutants on freshwater communities.

The authors used Grammarly to check spelling and grammar in the preparation of this manuscript.

Additional Information and Declarations

Competing Interests

Author Contributions

Data Availability

The authors declare there are no competing interests.

Rachel Becker conceived and designed the experiments, performed the experiments, analyzed the data, prepared figures and/or tables, authored or reviewed drafts of the article, and approved the final draft.

Elizabeth Hamman conceived and designed the experiments, analyzed the data, prepared figures and/or tables, authored or reviewed drafts of the article, and approved the final draft.

The following information was supplied regarding data availability:

The data and code used for analysis are available at Zenodo: eahamman. (2025). eahamman/Physa_Naproxen: Effects of naproxen sodium on Physa spp. (v1.0.0). Zenodo. https://doi.org/10.5281/zenodo.15529652.

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
