# Peer review of "The effects of naproxen sodium on the growth, reproduction, survival, and feeding of a freshwater pond snail"

_PeerJ, doi:10.7717/peerj.20163_

## Round 0.1 · original submission · Major Revisions

Three recognized experts have evaluated your submisson and identified a number of points that require a substantial revision of your manuscript. The most important aspects are (1) the lack of analytical verifications of nominal naproxen concentrations during the experiments, (2) that a more detailed description of materials and methods is required, and (3) the lack of information on background concentrations of naproxen in the environment. In addition, you should consider hypotheses that you want to prove with your experiments, and provide in the manuscript the link to the repository containing your raw data so that reviewers can access it.

On the other hand, the reviewers have underlined the importance of your study, so I hope that their criticisms will allow you to make a substantial revision of the manuscript, which is a prerequisite for its acceptance.

Reviewer 1 ·

Basic reporting

This study appears to provide new information on the effects of naproxen in freshwater snails, which has not previously been published. The manuscript should be carefully proofread by the authors as it contains some errors.

My first comment is that there should be a space between the value and the unit throughout the manuscript (e.g. 10 µg/L).

Introduction
L. 54: 2022 should not appear here
L.77 : triclosan is not a pharmaceutical product ; and it should not be written with a capital letter

Figures: Statistically significant differences should be indicated by asterisks (*p<0.05 vs control) or letters.

Experimental design

The experimental design is unclear. The material and methods are the weak point of the manuscript. The critical point is the lack of measurements of naproxen in water to confirm exposure concentrations, which are essential. In addition, the method followed by the authors is not precisely described and a lot of information is missing. However, the paragraph on statistical tests is complete and well argued.
The order in which the information is provided is confused; information on the provenance of the snails should appear before the experimental design. I suggest adding subheadings in this section to improve clarity.

L. 85. Information on chemical products (where naproxen sodium comes from) is missing.

L. 87 It is not clear, was each snail alone in a container? the material of the container that can influence the adsorption of the contaminant on the walls (glass is preferable to plastic) is missing. In addition, if the individuals were alone, it needs to be clarified what is meant by measuring the number of eggs laid. This would mean that reproduction took place before exposure to naproxen.

L.89 there is an error on the unit (11.4 µg/L) and on the source. The justification for the environmental values needs to be reviewed, as the lowest concentration corresponds to 10 times the highest concentration of naproxen found in the aquatic environment.
Świacka, K., Michnowska, A., Maculewicz, J., Caban, M., Smolarz, K., 2020. Toxic effects of NSAIDs in non-target species: a review from the perspective of the aquatic environment. Environmental Pollution 115891. https://doi.org/10.1016/j.envpol.2020.115891

L.93 information on acclimation time prior to experimentation is missing

L.97 the way in which the exposure was carried out is not clear, the concentration of the stock solutions is missing

L.99 Missing paragraph on how feeding was assessed

Validity of the findings

I suggest to also add titles to the results, and to follow the same order as in the materials and method to improve clarity.

L.120 ‘final 3 weeks’ is not clear, it is explained that the test lasts a month, so normally 4 weeks.

L.125 it would have been interesting to count the number of eggs in each egg sac

L. 134 The mortality rate of control snails is high (30%) compared with the literature, which must be justified.

L. 137 ‘in reducing growth’ should not be included in this paragraph. There seems to be a mistake.

L. 143 replace ‘0000µg/L’ with ‘10000 µg/L ‘.

L. 153 There is a comma instead of a full stop after ‘(Figure 1B)’.

L. 160 which pollutants

L. 193 this conclusion was not reached in the discussion. Indeed, the effects observed on Physa spp. in this manuscript have not been compared with other NSAIDs, with regard to the literature.

Additional comments

Although the subject covered in this article seems relevant in the light of the literature, the lack of data on materials and methods compromises the quality of the study. My biggest concern is the lack of measurement of naproxen in water. Without this data, or a clear justification of why it was not possible, I do not recommend publishing this article.
Furthermore, in order to refine as much as possible the quantity of organisms used in in vivo studies, it would be advisable to study as many parameters as possible during exposure. Here, little information was collected during the studies (for example, the number of eggs in each egg sac was not counted), which gives a low number of results. However, I appreciate the fact that the data and the code for the statistical tests have been made available on Github.
I recommend a major revision of this manuscript.

Reviewer 2 ·

Basic reporting

The Effects of Naproxen Sodium on the Growth, Reproduction, Survival, and Feeding of a Freshwater Pond Snail
Abstract
Line 19-20: Be consistent with your formatting. Provide a space between number and unit: 100 μg/L instead of 100μg/L.
Line 21: Reorganize the order of variables (growth, reproduction, survival, feeding) to match their sequence in the Title, Methods, Results, and Discussion sections.
Introduction
• You are using naproxen and naproxen sodium interchangeably. Pick one or say somewhere that you will use naproxen for naproxen sodium or if the studies focus on one and not the other, rewrite for naproxen sodium studies and naproxen studies. As written, I understand they are the same.
• You use common names or animals throughout the manuscript. For example, you say in line 64 the multiple effects on fish organs, provide information to which fish.
• What concentration is naproxen detected in freshwaters specifically in the US? Provide background on commonly detected concentrations (this will help support your choice of concentrations). How about maximum concentration of naproxen?
• Provide relevant studies that use Physids for ecotoxicology.
• Provide hypotheses
Line 34: Revise to "Insufficient wastewater treatment methods cause chemicals to enter the natural environment."
Line 43: Add citations to support the statement about aquatic ecosystem impacts.
Line 44 – 48: Provide the scientific names
Line 49 – 50: Rewrite for a simpler and easier to understand statement.
Line 56: Use the scientific names and the common name.
Line 60: You introduce naproxen sodium, and before is just naproxen. Provide information of why in the introduction
Lines 69-75: Provide additional references to support the claim that naproxen is more environmentally friendly than ibuprofen or diclofenac.
Line 71: Provide an explanation for demographic rates or an i.e. demographic rates (i.e. mortality, hatching rate…etc.)
Material and Methods
• Where did you get your naproxen? Were the concentrations analytical confirmed or are they nominal? What did the snails eat?
• How did you measure snail length? Caliper? ImageJ?
• How did you measure feeding?
• You measured number of egg sacs. Why didn’t you also count the number of eggs per egg sac, provide a reasoning.
Line 88: Clarify the sample size (n) of snails used in each treatment.
Line 89: support of environmentally relevant concentrations should be introduced in the introduction. Concentration of 11.4 mg/L is likely from hospital or manufacturing effluent. Please specify.
Line 94: Instead of mason jars, say glass jars. International readers might not know what a mason jar is.
Line 95: Synthetic pond water was made using EPA protocol? Specify in your methods. Provide calcium concentration
Line 96: Provide manufacturer details and describe the type of food provided to snails.
Line 98: Indicate how snail measurements were conducted (e.g., software like ImageJ or manual methods).
Results
• Include a table, summarizing the results.
• Throughout the paper, use 2 or 3 significant figures, including statistical results.
Line 141: “Snails ate…” how did you measure snails feeding. How did you analyze it?
Lines 142-143: Simplify phrasing, e.g., "At the highest concentration of naproxen..."
Figures: Include if there were significant effects or not.
Discussion
Line 151: why is naproxen preferable than other NSAIDs?
Line 156 – 163: in addition to hatching rate, number of eggs per egg sac could have been altered by naproxen.
Lines 164-165: Improve transitions between the impacts on Physa spp. and broader ecological implications.
Line 169: Be specific on floral community, Physa consumes biofilm and detritus. So in addition to ecosystem structure of biofilm communities, it can affect nutrient cycles as a detritivore.
Line 172 – 174: be more descriptive on how Physa can affect each one
Line 192: Expand on comparisons to studies like Naslund et al. 2020. Discuss how these findings align or differ.

Experimental design

.

Validity of the findings

.

Annotated reviews are not available for download in order to protect the identity of reviewers who chose to remain anonymous.

·

Basic reporting

I could not find raw data associated with this submission. Please provide raw data.

This interesting and well-written manuscript addresses an important research gap in our understanding of effects of NSAID pollution on freshwater organisms. The main weakness of the manuscript at present is that it does not provide enough information to fully evaluate the study. In addition, the interpretation of results in the discussion could be improved. Specific points are enumerated in the sections below.

Experimental design

More description is needed for methods and statistical analyses. See comments below.

1) Use of generalized linear mixed models is a good approach for these data and experimental design. However, not enough information is reported to evaluate validity of statistical approach.

1a) Please clarify which models included a quadratic term, how it was incorporated, and why. Was this time*time or something else? How as was the model for compared with and without the quadratic term? AIC values? Methods only mention a quadratic term for growth, while results only mention it for reproduction; was it in both or just one?

1b) With the poisson and binomial models, was overdispersion considered and, if necessary, corrected for?

1c) In the methods, please clarify what treatments are compared with each contrast in the survival analysis. For example, clearly state that second contrast described compares 100 vs 1000 and 10000. It is also not clear why, for this analysis, you chose to calculate contrasts rather than a different post-hoc approach like pairwise comparisons, as with the other variables.

2) A statistical approach that incorporates temporal autocorrelation would be more appropriate and could increase statistical power.

2a) The decrease through time observed for growth and reproduction suggests that observations from a particular individual may be more similar when closer together in time. Temporal autocorrelation could be incorporated, for example, by treating time as a continuous variable and using an autoregressive error structure for modeling the within-subjects error.

3) Please include more details in the methods.

3a) Provide the contents of all solutions, including the synthetic pond water and the naproxen solutions. For example, was the water tap water, well water, etc.? Was it filtered, sterilized, and/or treated with a conditioning agent? Were solutes added, and if so, in what concentration? Where were the solutes obtained? Was anything else added? What were the “stock solutions”? Where was the naproxen sodium obtained/who was the manufacturer?

3b) Please describe what snails were fed, and how much they were provided at each feeding. How was feeding assessed? Why was feeding treated as a binary variable?

3c) Please describe how snails were measured (i.e., with calipers, ruler, or by assessing landmarks on a photo).

3d) Were snails randomly assigned to treatments? Were jars randomly assigned within the physical layout in the laboratory (i.e., in a randomized block or fully randomized design)?

Validity of the findings

More details on statistical results are needed. The interpretation of results could be improved. See comments below.

4) Not enough information is reported to evaluate validity of statistical results.

4a) Please clarify what statistical tests are being reported by the p-values. Are these from post-hoc tests comparing each naproxen treatment with the no-naproxen control?

4b) Please also report statistical results for main effect of treatment and the overall time x treatment effect (i.e., considering all treatments together) as well as the quadratic term. Report test statistic values and degrees of freedom as well as p-values.

5) The manuscript implies that only the lowest concentration is environmentally relevant, but the evidence provided is unclear. If the lowest concentration is environmentally relevant, then the effect of naproxen on survival at this concentration is potentially more important than recognized in the discussion. If 100 micrograms/L is environmentally relevant, then the finding of a two-fold reduction in survival at this concentration could be very important ecologically. The abstract and discussion as currently written tend to deemphasize this point and instead focus on the more extreme levels that were tested.

5a) Please clarify the reasoning for your choice of concentrations. You mention previously reported environmental concentrations of 11.4 mg/L =11,000 micrograms/L, which is similar to your most extreme treatment. Is there a typo with units, or is 11.4 mg/L an unusually extreme environmental level? If the latter, it would be helpful to provide more information about more typically observed environmental levels. My reading of the manuscript was that your 100 microgram/L treatment represented an “observed” or environmental concentration whereas the other two were more extreme. This point is particularly important for how your results are to be interpreted.

5b) If 10,000 micrograms is indeed an order of magnitude higher than previously tested levels and two orders of magnitude higher than environmental levels, it would be helpful to provide a justification for testing such a high level. Are there scenarios under which organisms could be exposed to such a high level (e.g., in the case of direct release into the environment) or does testing this extreme concentration potentially provide some insights into the effects of naproxen on the snails’ biology?

5c) Lines 156-157 – Expand the discussion of the survival result. It appears that survival at 100 micrograms/L was less than half that of the control at the end of the experiment. If a similar effect level occurred under field conditions, population size could be affected which could alter the community. Observing an effect of this magnitude at an environmental level is thus an important result, and its potential consequences should be explored more deeply. I suggest devoting a paragraph to this discussion and then beginning a new paragraph to discuss the reproduction results.

5d) Lines 193-195 – The statement that the effects observed in this study were “generally at a much higher concentration than are currently in aquatic ecosystems” seems at odds with finding a sizable decrease in survival at the lower, environmentally-relevant concentration. If your point is that this concentration, though observed, is uncommon, then you’ll need to provide more information about the range of concentrations observed in nature. On the other hand, if your point is that more extreme effects were observed at even higher concentrations, this does not mean that the effect on survival at the lower concentration is ecologically unimportant.

5e) Omit final line of abstract, which de-emphasizes the important finding that survival is affected at an environmentally-relevant level. A more satisfying ending would be to revise the penultimate sentence to focus on the effects on survival (rather than demographic rates in general).

5f) Line 168-169 – Clarify that you’re referring to reductions in grazing due to the effects of naproxen on snail survival, which were observed at environmentally-relevant levels, rather than the direct effects of naproxen on feeding rates, which were only observed at extreme concentrations.

5g) On a related note, in reporting and discussing results, it is a bit confusing to refer to “observed,” “elevated” and “extreme” concentrations without clearly defining how you are using these terms (e.g., lines 144, 156). Please either state clearly what treatment each term represents or be clearer about what concentrations you’re referring to in each case throughout the manuscript. Finally, “environmental” or “environmentally-relevant” may be more easily understood by the reader than “observed” to mean a level that has been reported in the environment.

Additional comments

6) Other minor comments/suggestions (in order of occurrence):

6a) Lines 30-31 – Consider revising “detected in natural environments and pollution documented across…” to “documented in natural environments across…”

6b) Line 50 – add “populations and” before “communities”

6c) Lines 52-54 – Justify why snails are a good indicator. Is this because of their ecological role, because their tolerance levels are often representative of those of other organisms, or something else? The reasoning that molluscs are highly represented in such studies does not necessarily mean that they’re good indicators, but more likely that they’re a convenient system for such tests.

6d) Lines 61, 72 – It would be helpful to move the point that naproxen has been documented in mollusc tissue to the previous paragraph (e.g., after the first sentence).

6e) Line 88 – change “replicants” to “replicates”

6f) Line 142 and 182 – change “Figure 1B” to “Figure 2”; line 183 – change “Figures 1 and 2” to “Figure 1”

6g) Line 143 – typo in reporting naproxen concentration (missing a 1)

6h) Lines 149-152 are unnecessary and can be omitted

6i) Line 159 – It’s also possible that naproxen could affect other reproductive variables, like egg size or number of embryos per mass.

6j) Line 164 – extra word between “freshwater” and “snails” – omit

6k) Lines 184-185 – Clarify that the effects of naproxen could also be overestimated in your experiment, as the stress of laboratory conditions could exacerbate the effects of naproxen.

6l) Lines 190-191 – This is an oversimplification. Some pharmaceuticals have a much greater tendency to bioaccumulate than others.

6m) In the Figure 1 caption, please include what the box and whisker plots represent (e.g., quartiles, range?).

6n) Also in Figure 1, the yellow dots in panel C are somewhat hard to see due to low contrast with the white background. Either use a more saturated color or outline each dot in black.

---

## Round 0.2 · Minor Revisions

We are almost there! The reviewers suggest a further changes. I agree that these are necessary to accept the manuscript for publication after the appropriate updates.

Reviewer 1 ·

Basic reporting

I thank the authors for their response to my previous remarks and for the corrections made to the manuscript, which have improved the overall quality of the article. Here some additional comments: in some cases, I was unable to find the corrections in the revised text (for example, regarding the duration of the acclimation period). I suggest that you indicate the line number of each correction directly in your response to the reviewers to facilitate verification.

General remark: Please be consistent in the notation of concentrations. Choose between the formats "1,000 and 10,000" or "1000 and 10000", and apply the chosen format uniformly throughout the manuscript.

Abstract: The mention of “feeding rates” is confusing in this context. Consider rephrasing as “the rate of snails fed”.

L.46: Add a comma between Daphnia magna and Thamnocephalus platyurus.
L.49: Correct the species name to Oryzias latipes.
L. 67: Add a space after "(Wolecki et al., 2019)".
L.69: Add a space after "rotifer".
L.73: Add a space after "2020)."
L.101: How long were the snails held before the start of the experiment? Please clarify.
L.107: What type of food was given to the snails? Please specify.
L.128–129: The sentence should be corrected for grammar. Suggested revision: "Only living snails were included in the analysis of growth, reproduction, and feeding."
L.130: Replace “oz” with "g" to maintain consistency with SI units.
L.204: Add a space between "in" and "zebrafish".
L.227: Add a space after "(Figure 2)".

Results: There is a formatting issue in the PDF: a square symbol appears before each citation. This should be corrected.

Experimental design

/

Validity of the findings

/

Reviewer 2 ·

Basic reporting

The authors have addressed the feedback provided by the three reviewers. However, specific research on Physa acuta and pharmaceutical exposure is still missing from the revised draft. For example, while Bernot’s work addresses locomotion, it focuses on ionic liquids and nanosilver, which differ from the pharmaceutical contaminants examined in this study. Other research have studied the effects of pharmaceuticals on Physa acuta, for example:

1. Effects of ibuprofen and microplastics on movement, growth and reproduction in the freshwater snail Physella acuta
2. Impact of acetaminophen and microplastic exposure on Physa acuta movement, growth, and reproduction
3. Effects of individual and combined pesticide commercial formulations exposure to egestion and movement of common freshwater snails, Physa acuta and Helisoma anceps

Experimental design

The methods and results have been clarified and described in much more depth.

Validity of the findings

Findings and results are valid

Additional comments

N/A

·

Basic reporting

Thank you for sharing raw data and providing link within the manuscript.

Overall, the revisions make important improvements to the manuscript. However, some of the revisions have introduced a lack of clarity, problems of organization, and awkward phrasing. Suggestions for improvement are noted below under Additional Comments.

Consider reorganizing the order of variables throughout the manuscript to lead with survival. I appreciate that you already reorganized for consistency in the order of response variables throughout manuscript. However, because survival is the most important result of the study and also, arguably, the most direct and ecologically important measure of the impact of naproxen on organismal fitness, I think your findings would be conveyed more strongly by reorganizing throughout to put survival before the other response variables.

Experimental design

Thank you for elaborating on the details of the experimental methods and the statistical analyses.

While it would be preferable to have analytically verified the naproxen concentrations, I am satisfied with the authors' presentation of nominal concentrations.

I still do not see a source for naproxen sodium. Please add this information.

Details of the calcium carbonate supplementation (i.e., what was the concentration of calcium carbonate in the solution, or was it added to saturation, or something else) must be provided. Please do your due diligence and track down the care sheet or some other record of how you added calcium carbonate.

Strictly speaking, it is not correct to state that naproxen and naproxen sodium are the same, particularly if concentrations are reported in ppm or mg/L because the sodium ion adds mass to the molecule that isn't there if the concentration of just naproxen were to be reported.

There is some lack of clarity about the acclimation conditions. See recommended revisions described below.

Validity of the findings

Solid work.

Additional comments

As mentioned above, in revising the manuscript, the authors have introduced some awkward phrasing, unclear organization, and other issues. These issues can be addressed with careful proofreading and thoughtful revision. See suggestions below:


The organization of the Introduction could be improved.

* For simplicity and clarity, in the intro, refer only to naproxen, rather than alternating between naproxen and naproxen sodium (e.g., lines 66, 73, 92, 94). (This is valid as naproxen is the active component of naproxen sodium, thus the effects observed in studies of naproxen sodium can be attributed to naproxen.) Also, move the info about naproxen from lines 39-41 to the methods section.

* Consolidate the background information on biological effects of naproxen by moving the information from lines 48-49 (naproxen effects on medaka) to the paragraph on naproxen (lines 68-76).

* Consolidate the information on molluscs by adding the information from lines 65-68 (nap relatively unstudied in molluscs) to the paragraph on molluscs as bioindicators (lines 56-64). Place this information after the paragraph ending on line 81 (lack of info on nap effects on freshwater organisms).


Additional clarification and reorganization of the Experimental Setup and Experimental Methods sections of the Materials and Methods is required.

* Describe the acclimation conditions. Move lines 111-112 to after sentence ending on line 101. If the glass jars, Holtfreters' solution, temp control, feeding and water change regime applied during acclimation as well as during the experiment, then revise line 101 to state "During the acclimation period and the experiment". If not, use this space to describe the housing conditions, water, etc. for the acclimation period and then describe the conditions that were used only in the experiment later (see below) and perhaps rename this section to "Animals".

* The feeding procedure needs to be described before mentioning food manufacturer directions. Thus, move the info about the food wafer from lines 130-131 to line 107. Lines 130-131 should only retain information about the experimental feeding procedure that was not already described for the acclimation period (e.g., if the amount and/or duration was controlled during the experiment but not during acclimation).

* Line 108 - omit "at each nominal concentration" - this doesn't make sense until after you've mentioned the concentrations in the next section.

* Move information about naproxen sodium and its stability to the Experimental Methods section.
**Line 115 - Change "naproxen sodium" to "naproxen".
**Include naproxen source here (line 118).
**Move lines 39-40 to line 118. Replace the statement that the two terms are often used interchangeably in the literature (line 41 - see above, this is not true, strictly speaking) with a statement that naproxen sodium is often used to test the biological effects of naproxen in non-target organisms.
**Line 119 - change "these levels" to "our experimental levels"
**Consolidate lines 108-109 with the information on line 122 and specify a time period over which naproxen is relatively stable -- it would be helpful to report a half-life here.

* Move lines 110-111 (randomization procedures) to a new paragraph after line 122. Perhaps remove number of replicates from line 118 and include here to round out this paragraph. Also mention the total number of snails (40) in the experiment and describe any conditions (e.g., housing, temp, light, feeding) that differed from the acclimation conditions.


The organization of the Discussion could also be improved.

* Omit lines 197-198 or combine this with information in the paragraph starting on line 207 to consolidate the discussion of grazing.

* Move lines 210-211 to the beginning of this paragraph to make the main point clearly.

* Can the information on lines 216-220 be condensed? I don't think this level of detail is necessary to make your point. For example, you could sum this all up by adding to the end of line 215, "such as parasite presence (ref), predator presence (ref), sediment loading (ref) and water flow (ref)"

* Move paragraph on reproduction (lines 199-206) to after line 224.


Proofread carefully throughout for:
* missing spaces (e.g., lines 67, 73, and elsewhere)
* special characters not translated (e.g., lines 152, 155, etc.)
* run-on sentences (e.g., lines 56-59)
* non-parallel construction (e.g., lines 178, 186)
* incomplete sentences/sentence fragments (line 192-194)
* other typos (e.g., lines 232-233


Other suggestions:

Line 29: change to "600 pharmaceuticals documented"

Lines 67-68: Can you include a statement that, to your knowledge, the effects of naproxen have not previously been studied in molluscs?

Line 81: Add "such as freshwater molluscs" to the end of this sentence.

Line 128-129: Move this sentence to the Statistical analyses section, to line 136

Line 139: Was the week x treatment interaction tested for each variable? If so, state that clearly. If not, state which models retained this interaction.

Lines 156-157, 162-163: Clarify that these are the results for the week x treatment interaction.

Line 176: Change "fed 41% less" to "fed significantly less"

Line 178: Change "nor between" to "while there was no significant difference between"

Line 178: Also include the results for the pairwise comparisons between 0 and 100 and 0 and 1000 ug/L.

Line 187: Change "but reduced survival" to a more complete thought such as, "but naproxen did reduce survival" (Or, better yet, reorganize throughout to put survival first, which would allow for less awkward phrasing of this sentence.)

Line 189: Change "observed levels of naproxen and higher" to "all tested levels of naproxen" or "a nominal level of 100 ug/L and higher"

Line 192: add change "members" to "other members"

* Line 207: change "snails have" to "snails also have"

* Line 199: omit the word "Additionally" from the beginning of this sentence.

* Line 203: change "as well as" to "and"

* Line 232: change "to do" to "due to"

* Line 232-233: change "the laboratory" to "naproxen"

* Line 242: change "their effects" to "their ecotoxicity"

* Line 249: omit the word "only"

* Figure 1 caption: the word "green" appears an extra time - proofread carefully

---

## Round 0.3 · Minor Revisions

I am sorry to inform you that another round of revisions is necessary. Reviewer 1, in particular, has identified several additional points that require revision. I also ask you to take the reviewers' comments seriously, or - if you disagree with their suggestions - to clearly explain in your statement why you are deviating from them.

Reviewer 1 ·

Basic reporting

Thank you for addressing the comments. However, as mentioned in my previous feedback, you should indicate the lines where corrections were made in your response letter. There are still several typos in the text, which needs to be carefully proofread.

Here are some comments (line numbers refer to the manuscript without tracked changes):

L.74: This statement is incorrect. According to the cited source, exposure to diclofenac actually increased L. stagnalis growth. In another study, growth was reduced under some conditions, but this was observed after multigenerational exposure:
Multigenerational responses in the Lymnaea stagnalis freshwater gastropod exposed to diclofenac at environmental concentrations https://doi.org/10.1016/j.aquatox.2022.106266

Experimental methods: how often was the exposure solution renewed? Was the exposure prepared from a stock solution, and at what concentration?

L.118: please add some references where naproxen sodium was used in similar experiments.

L. 119: even though your tested concentrations are below environmental levels, you should include information on environmental concentrations of naproxen in this paragraph or in the introduction.

L.130: The snails can self-fertilize but they were four per jar, they can also cross-fertilize; please clarify.

L.133: Specify at what point during the exposure period was snail feeding measured.

Discussion: At the beginning of the discussion, you state that naproxen may be less toxic to aquatic organisms than other NSAIDs such as ibuprofen or diclofenac. I suggest you discuss your findings in light of results obtained on other freshwater molluscs exposed to these compounds. For instance, another reviewer provided a reference (Elias et al., 2025) showing effect of ibuprofen on Physella acuta at similar traits to those assessed in your study (egg production), and at lower concentration. This is also possible with other studies cited in your introduction on NSAIDs and mollusc exposure.
However, this study is not cited in your manuscript, could you clarify why?
Effects of ibuprofen and microplastics on movement, growth and reproduction in the freshwater snail Physella acuta https://doi.org/10.3389/fenvs.2024.1514062
You could also compare your results with other studies on NSAIDs and mollusc exposure that are already cited in your introduction, to strengthen your discussion.

L.222-223: Please delete the part of the sentence referring to zebrafish (“and antibiotics in zebrafish (Bielen et al., 2017)”) as it is not relevant to this paragraph.

Experimental design

/

Validity of the findings

/

Additional comments

Minor comments:
L.18: Replace “, on freshwater molluscs” with “and freshwater molluscs”.
L.22: delete “sodium”
L. 54: add a comma after (Lucero et al., 2015)
L110: Please delete the extra full stops (“.” instead of “…”).
Results: Please ensure consistent formatting, for example whether there is a space between “χ²” and “(1)” and around “=”.
L. 212: add a space between “as” and “parasite”
L. 241: add a point after pharmaceutical
Figure 1 and 2: There is no need to restate the color coding (“Green indicates snails not exposed to naproxen, teal represents snails exposed to 100 µg/L, blue represents snails exposed to 1,000 µg/L of naproxen and purple represents snails exposed to 10,000 µg/L.” ) in the figure captions, as this is already provided in the figure legends.

Reviewer 2 ·

Basic reporting

No comment

Experimental design

No comment

Validity of the findings

No comment

Additional comments

No comment

·

Basic reporting

Thanks for the revisions. Much improved!

Experimental design

In the methods (lines 140-153 in the tracked changes version), please specify whether the experimental naproxen solutions were made in dechlorinated tap water, in Holtfreter's soltuion, or in something else.

Validity of the findings

Nice work!

Additional comments

Just a few suggested changes to wording. In all cases, line numbers cited below are in the tracked changes version.

line 56 - revise the phrase "fish medaka fish" so that the word "fish" only occurs once

line 64 - change "posit" to "posit that"

line 245 - with the new topic sentence for this paragraph, the word "also" is no longer needed here

lines 254-257 - double check to make sure this list includes all necessary commas. I think one may be missing, but I'm not 100% sure without seeing a clean (non-tracked-changes) version, which I did not see as part of the provided files.

line 264 - change "the effect" to "an effect"

line 280 - change "of the laboratory were exacerbated by naproxen conditions" to "of naproxen were exacerbated by laboratory conditions"

line 289 - change "diclofenac it" to "diclofenac"

---

## Round 0.4 · accepted · Accept

Thank you for the thorough revision of the manuscript. I hereby certify that you have adequately taken into account the reviewers' comments and improved the manuscript accordingly. Based on my assessment as an Academic Editor, your manuscript is now ready for publication.